# Rethinking Fairness Representation in Multi-Task Learning: a Performance-Informed Variance Reduction Approach

## Abstract

Multi-task learning (MTL) can leverage shared knowledge across tasks to improve data efficiency and generalization performance, and has been applied in various scenarios. However, task imbalance remains a major challenge for existing MTL methods. While the prior works have attempted to mitigate inter-task unfairness through loss-based and gradient-based strategies, they still exhibit imbalanced performance across tasks on common benchmarks. This key observation motivates us to consider performance-level information as an explicit fairness indicator, which can more accurately reflect the current optimization status of each task, and accordingly help to adjust the gradient aggregation process. Specifically, we utilize the performance variance among tasks as the fairness indicator and introduce a dynamic weighting strategy to gradually reduce the performance variance. Based on this, we propose PIVRG, a novel performance-informed variance reduction gradient aggregation approach. Extensive experiments show that PIVRG achieves state-of-the-art performance across various benchmarks, spanning both supervised learning and reinforcement learning tasks with task numbers ranging from 2 to 40. Results from the ablation study also show that our approach can be integrated into existing methods, significantly enhancing their performance while reducing the variance in task performance, thus achieving fairer optimization.

## 1 Introduction

Multi-task learning (MTL) (Caruana, 1997; Ruder, 2017; Zhang & Yang, 2021) is an approach where a single model is trained to solve multiple tasks simultaneously. This paradigm allows tasks to share information and representations, which can enhance the generalization capabilities of the model and improve performance across tasks (Baxter, 2000; Standley et al., 2020; Navon et al., 2020). MTL is especially beneficial in scenarios where computational resources are limited, as it reduces the need for separate models for each task. Its applications span a variety of domains, such as computer vision (Achituve et al., 2021; Zheng et al., 2023; Liu et al., 2019), natural language processing (Chen et al., 2024; Liu et al., 2017; Pilault et al., 2020), and robotics (Devin et al., 2017; Xiong et al., 2023). Despite its advantages, MTL faces one major significant challenge: task imbalance, which describes the phenomenon where some tasks dominate the learning process while others are under-optimized, leading to degraded overall performance. Overcoming this challenge requires careful design of optimization strategies to ensure that all tasks benefit equally from the shared model.

To address these issues, previous works have focused on two primary approaches: loss-based and gradient-based methods (Liu et al., 2021b; Senushkin et al., 2023). Loss-based methods aggregate the losses of different tasks by adjusting the loss scales, then backpropagate the total loss to compute gradients for shared parameters. These methods attempt to reflect the optimization status of each task through their respective losses. However, since tasks often use different loss functions (e.g., cross-entropy for classification and L1 loss for regression), their scales can differ significantly, making it hard to compare or balance them directly. Various techniques have been proposed to normalize these losses, such as linear scaling, logarithmic scaling, and polynomial scaling, but the fundamental issue of differing loss magnitudes remains. On the other hand, gradient-based methods compute the gradient for each task separately and then aggregate these gradients using various algorithms to

produce a final update. Although these methods ensure that the gradient of each task is considered, relying on the gradients to represent optimization fairness can be misleading at certain stages of the optimization process. For instance, the gradient approaches zero at a local minimum, while the optimization state may still be suboptimal. Additionally, these methods typically aim to equalize the task gradients at the shared layers (Chen et al., 2018; Liu et al., 2021b), but can not guarantee the balance in the training progress because the difficulty of tasks may differ. Easier tasks may converge quickly, while difficult tasks require more time to optimize (Guo et al., 2018). As a result, considering only gradients can overlook differences in task difficulty, making it insufficient to ensure balanced optimization across all tasks.

A clear illustration of the issues above can be observed in the widely used NYUv2 benchmark (Silberman et al., 2012), which involves three tasks: segmentation, depth estimation, and surface normal prediction. While recent MTL methods have demonstrated improvements over single-task learning (STL), as evidenced by negative average performance drops $\Delta m$ (Navon et al., 2022; Liu et al., 2024; Ban & Ji, 2024), their experiments show that these methods outperform STL primarily on the segmentation and depth tasks. However, their performance on the surface normal task consistently lags behind STL, leading to a substantial variance in the performance drop across tasks. This variance contradicts the original goal of MTL, which aims to achieve balanced optimization across all tasks. This key observation prompts us to rethink: *Is information from loss-level and gradient-level metrics sufficient to represent fairness in multi-task optimization?* In our view, performance-level information should also be considered, and $\Delta m$ is a good choice for this purpose. On one hand, $\Delta m$ serves as the final performance metric we ultimately compare, providing a direct and definitive reflection of each task's optimization status. On the other hand, $\Delta m$ for each task reflects its relative performance drop compared to its respective STL baseline, leading to an invariant scale across tasks. This property allows for a direct comparison of optimization progress and provides a guideline for promoting fairness in the optimization process. While this observation may seem intuitive, it is precisely the aspect that has been overlooked by most previous MTL methods.

Building upon the above motivation that considers performance-level fairness, we employ the performance variance across tasks as an indicator and implement a dynamic weighting approach aimed at progressively decreasing this performance variance. This enhances the generalization and robustness of the shared representations, reducing excessive performance discrepancies between tasks. In summary, our contributions can be outlined as follows:

**1). We rethink the fairness representation in MTL optimization and suggest incorporating performance-level information as a prior.** Based on the common task imbalance issues observed in the NYUv2 benchmark, we argue that loss-level and gradient-level information is insufficient to capture fairness in MTL. Instead, performance-level information should be considered to reflect the varying difficulty levels across different tasks.

**2). We propose PIVRG, a novel performance-informed variance reduction gradient aggregation approach.** Specifically, we utilize the performance variance among tasks as a fairness indicator and introduce a dynamic weighting strategy, which serves as a regularization mechanism balancing the performance drop across different tasks during MTL. Both theoretical analysis and experimental results demonstrate that our method not only converges to the Pareto stationary point but also achieves superior performance.

**3). Extensive experiments demonstrate that PIVRG achieves state-of-the-art performance across various benchmarks.** Notably, on the NYUv2 benchmark, unlike previous methods that consistently underperform compared to STL on the surface normal task, PIVRG surpasses STL across all three tasks and achieves the best overall performance drop. Moreover, on the Cityscapes and CelebA benchmarks, PIVRG is the first to achieve a negative performance drop, meaning it surpasses STL in average performance for the first time. On the more challenging QM9 benchmark, PIVRG reduces the average performance drop by over 20%.

**4). The proposed performance-informed dynamic weighting strategy is orthogonal to existing approaches, making it possible to integrate with these methods.** Experimental results demonstrate that incorporating our dynamic weighting strategy not only significantly improves the overall performance of these methods but also substantially reduces the performance variance across tasks, leading to a more balanced optimization. This further validates the potential of our approach.

## 2 RELATED WORK

**Loss-based MTL approaches.** These methods reweight task-specific losses with loss-level information. A key advantage of loss-based methods is their efficiency, as they only require backpropagation on the aggregated loss, reducing computational overhead compared to handling each task individually (Liu et al., 2024). These approaches include operations on the scale of the loss, such as Linear Scalarization (LS), which minimizes the sum of task losses, and Scale-Invariant (SI), which reduces the sum of logarithmic losses. Additionally, various approaches for handling task weights have been proposed, including using homoscedastic uncertainty weighting (Kendall et al., 2018b), task prioritization (Guo et al., 2018), dynamic weight averaging (Liu et al., 2019), self-paced learning (Murugesan & Carbonell, 2017), geometric loss (Chennupati et al., 2019), random loss weighting (Lin et al., 2021b), impartial loss weighting (Liu et al., 2021b) and fast adaptive optimization (Liu et al., 2024). Although loss-oriented methods are more computationally efficient, they often underperform gradient-oriented ones in most multi-task benchmarks.

**Gradient-based MTL approaches.** These methods address the task-balancing problem by fully leveraging the gradient information of the shared network across different tasks. Several studies have reported notable performance improvements using techniques such as Pareto optimal solutions (Sener & Koltun, 2018), gradient normalization (Chen et al., 2018), projecting gradient conflicts (Yu et al., 2020a), gradient sign dropout (Chen et al., 2020), impartial gradient weighting (Liu et al., 2021b), conflict-averse gradients (Liu et al., 2021a), independent gradient alignment (Senushkin et al., 2023), and Bayesian uncertainty gradients (Achituve et al., 2024). Recent works by Navon et al. (2022) and Ban & Ji (2024) employ the Nash bargaining solution and fair resource allocation respectively to address the gradient aggregation problem. Their utility functions are primarily based on the first-order Taylor expansion of the loss, thereby incorporating loss-level information. Following Navon et al. (2022) and Ban & Ji (2024), our proposed PIVRG method is also a gradient-based approach. However, it not only incorporates loss-level information but also introduces higher-order insights from the performance-level.

## 3 METHOD

### 3.1 PRELIMINARIES

**Pareto Optimality.** Optimization in MTL can be understood as a specific instance of multi-objective optimization (MOO). For a set of objective functions $\ell_1, \cdots, \ell_k$, the quality of a solution $x$ is determined by the vector of its corresponding objective values, i.e., $(\ell_1(x), \cdots, \ell_k(x))$. A key characteristic of MOO is the absence of a natural linear ordering for such vectors, meaning that solutions are not always directly comparable, and thus no single optimal solution exists.

We define a solution $x$ as dominating another solution $x'$ if it is strictly better in at least one objective while being no worse in all others. A solution that is not dominated by any other solution is termed Pareto optimal, and the set of all such solutions forms the Pareto front. In the case of non-convex problems, a solution is considered locally Pareto optimal if it is Pareto optimal within a neighborhood around it. Furthermore, a solution is called Pareto stationary when there exists a convex combination of the gradients at that point that equals zero, which is a necessary condition for Pareto optimality.

**Multi-Task Optimization Objectives.** One of the most crucial distinctions between different MTL methods lies in their choice of optimization objectives. A traditional approach is to minimize the average loss across all tasks:

$$\min_\theta \left\{ \mathcal{L}(\theta) := \frac{1}{k} \sum_{i=1}^{k} \ell_i(\theta) \right\}, \tag{1}$$

where $\theta \in \mathbb{R}^n$ is the parameter shared across tasks. Directly optimizing Eq. 1 can lead to significant under-optimization for certain tasks and this is often caused by the varying scales of the different loss functions as discussed in Sec.1. Gradient-based methods typically propose an aggregation algorithm $\mathcal{A}$ (e.g., conflict projection (Yu et al., 2020a), cosine similarity balancing (Liu et al., 2021b)), which solves an optimization problem $\mathcal{A}(g_1, g_2, \cdots, g_k)$ to obtain the update direction $d$. Recent works (Navon et al., 2022; Ban & Ji, 2024) represent updates at each iteration as $\theta_{t+1} = \theta_t - \eta d$, where

$\eta$ is the current step size and $d$ is the computed update direction. Considering a first-order Taylor expansion $\ell_i(\theta_{t+1}) - \ell_i(\theta_t) \approx -\eta g_i^\top d$, they interpret $g_i^\top d$ as the utility of task $i$ at the current step, thus taking loss-level information into account.

In this paper, we also consider $g_i^\top d$ as the utility of task $i$ at the current step. However, unlike Nash-MTL (Navon et al., 2022), which aims to maximize the sum of the log-utilities, inspired by utility balancing and risk aversion principles in game theory (Pratt, 1978; Chen & Hooker, 2021), we propose to minimize the mean of the inverse utilities for each task:

$$\arg\min_d \frac{1}{k} \sum_{i=1}^{k} \frac{1}{g_i^\top d} \qquad \text{s.t.} \quad g_i^\top d > 0, \forall i. \tag{2}$$

This approach emphasizes tasks with lower utilities, thereby preventing tasks with high utilities from dominating the optimization process. In fact, from another perspective, $g_i^\top d$, as an approximation of the change in loss, can be viewed as the current optimization speed of task $i$, while $1/g_i^\top d$ can be interpreted as the number of steps required for unit improvement. The objective in Eq. 2 essentially minimizes the average number of steps needed for unit optimization across tasks.

## 3.2 Performance-Informed Weighting Strategy

To mitigate the task imbalance issue mentioned in Sec. 1, we propose incorporating performance-level information $\boldsymbol{\Delta m} = (\Delta m_1, \Delta m_2, \cdots, \Delta m_k)^\top$ to account for the varying difficulties across tasks. Specifically, for each task $i$, following previous works (Sener & Koltun, 2018; Navon et al., 2022; Liu et al., 2024), we define the performance drop $\Delta m_i$ as:

$$\Delta m_i = (-1)^{\delta_i}(M_{m,i} - M_{b,i})/M_{b,i} \times 100, \tag{3}$$

where $M_{b,i}$ is the value of metric $M_i$ obtained by the STL baseline and $M_{m,i}$ denotes the value from the compared MTL method. $\delta_i = 1$ if a higher value is better for the metric $M_i$ and 0 otherwise. This ratio quantifies the relative degradation of performance when tasks are optimized jointly.

Moreover, $\Delta m = \frac{1}{k} \sum_{i=1}^{k} \Delta m_i$ is a metric reflecting the overall performance of the MTL method across tasks. While reducing the average performance drop $\Delta m$ is the goal of all MTL methods, we utilize the performance variance $\text{Var}[\Delta m_i]$ among tasks as a fairness indicator and consider it as a potential optimization target for ensuring fairness. Note that $\Delta m_i$ is not a random variable and we just use $\text{Var}[\cdot]$ as a formal notation to represent the variance of performance drop.

Since $\Delta m$ represents actual performance and lacks gradients for backpropagation, it cannot be directly optimized for variance reduction without further assumptions. Thus, we introduce dynamic weights $\boldsymbol{\omega} = (\omega_1, \omega_2, \ldots, \omega_k)^\top \in \mathbb{R}^k$ as regularizers to guide the optimization process indirectly. Specifically, we rewrite the original objective as:

$$\arg\min_d \frac{1}{k} \sum_{i=1}^{k} \frac{\omega_i}{g_i^\top d} \qquad \text{s.t.} \quad g_i^\top d > 0, \forall i. \tag{4}$$

In Eq. 4, the choice of $\boldsymbol{\omega}$ is crucial. We aim for $\boldsymbol{\omega}$ to reflect the current performance-level information of each task and to promote reducing $\text{Var}[\Delta m_i]$ during optimization. To this end, $\boldsymbol{\omega}$ should satisfy the following properties:

**Property 1.** $\omega_i$ *should be positively correlated with* $\Delta m_i$.

The objective in Eq. 2 minimizes the average number of steps required for unit optimization across tasks. However, as previously discussed, different tasks have varying difficulties, and more difficult tasks may require more steps at the same optimization step size. Without considering task difficulty, the objective in Eq. 2 might result in over-optimization of some tasks while others remain under-optimized, thus maintaining task imbalance. By modifying the weights $\omega_i$ to be positively correlated with $\Delta m_i$, the objective becomes aware of task difficulty. Eq. 4 can still be seen as minimizing the average number of steps across tasks, but for tasks with a larger $\Delta m_i$ (i.e., less optimized tasks), we expect $\omega_i > 1$ to encourage more aggressive optimization. Conversely, for tasks with smaller $\Delta m_i$, we expect $\omega_i < 1$ to slow down the optimization for that task. This dynamic adjustment ensures a more balanced performance across tasks and helps mitigate excessive variance.

**Property 2.** $\mathbf{1}^\top \boldsymbol{\omega} = k$.

This ensures alignment with the original objective in Eq. 2 without the weight. In the unweighted case, $\boldsymbol{\omega} = (1, 1, ..., 1)^\top$ satisfies $\mathbf{1}^\top \boldsymbol{\omega} = k$. Our weighting strategy dynamically adjusts this $k$ from a fixed mean to a more flexible distribution, and adds correlation to the weights of different tasks.

**Property 3.** $\boldsymbol{\omega}$ *is bounded, i.e.,* $\omega_i \in [\underline{\omega}, \overline{\omega}]$.

This constraint ensures that the weight does not become too extreme, for instance, preventing one task with poor performance from consuming all resources (especially during early training when $\text{Var}[\Delta m_i]$ might be large). In practice, we typically choose $\underline{\omega} \in [0.5, 0.8]$ and $\overline{\omega} \in [1.2, 2]$.

Property 2 is essentially a special case of normalization, making the commonly used softmax function a natural choice. Applying softmax to $\boldsymbol{\Delta m}$ also ensures the positive correlation required by Property 1. However, we observe that direct normalization makes it challenging to enforce $\omega_i \in [\underline{\omega}, \overline{\omega}]$. To address this, a simple yet effective idea is to adopt a variant of softmax with a temperature parameter, as follows:

$$\omega_i = \frac{k \cdot \exp(\Delta m_i / \tau)}{\sum_{j=1}^{k} \exp(\Delta m_j / \tau)} \tag{5}$$

where $\tau$ is the temperature parameter controlling the smoothness of the softmax output. We assert that by choosing an appropriate $\tau$, Property 3 can also be satisfied.

**Proposition 1.** *Let* $\Delta m^{max}$ *be the maximum value of* $\boldsymbol{\Delta m}$, *and* $\Delta m^{min}$ *be the minimum value. Define* $s = \min\left(\frac{1}{\underline{\omega}}, \overline{\omega}\right)$. *Then, for* $\tau > \frac{\Delta m^{max} - \Delta m^{min}}{\log s}$, *Property 3 is satisfied.*

The proof can be found in the Appendix. We also demonstrate that the fairness indicator and potential optimization target, $\text{Var}[\Delta m_i]$, can be approximated by the norm of $\boldsymbol{\omega}$, specifically $\boldsymbol{\omega}^\top \boldsymbol{\omega}$.

**Proposition 2.** *For* $\boldsymbol{\omega}$ *satisfying the three properties above, we have the following approximation:*

$$\text{Var}[\Delta m_i] \approx \frac{\tau^2}{k} \boldsymbol{\omega}^\top \boldsymbol{\omega} - \tau^2.$$

This implies $\text{Var}[\Delta m_i] \propto \boldsymbol{\omega}^\top \boldsymbol{\omega}$. The detailed proof can be found in the Appendix. The result in Fig. 1 shows that PIVRG outperforms other approaches and produces the lowest performance variance, indicating the capability of our method. Furthermore, the experimental results in Appendix B.3 demonstrate that both $\boldsymbol{\omega}^\top \boldsymbol{\omega}$ and $\text{Var}[\Delta m_i]$ decrease progressively throughout the optimization process, further confirming the effectiveness of our dynamic weights which serve as regularizers. After refining the definition of $\boldsymbol{\omega}$, the problem reduces to solving for the optimal $d$ in Eq. 4.

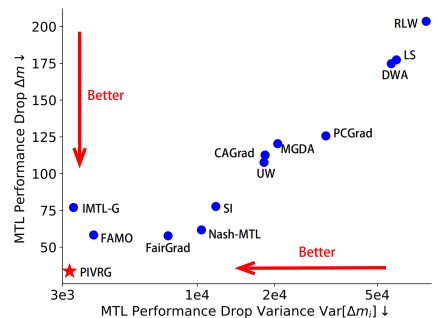

Figure 1: Experiment results about $\Delta m$ and $\text{Var}[\Delta m_i]$ on the QM9 dataset.

### 3.3 Deriving the Optimal Update Vector $d$

Given an MTL optimization problem and parameters $\theta$, we search for the update vector $d$ in the ball of radius $\epsilon$ centered around zero, $B_\epsilon$. First, we show that both the objective function and constraints are convex:

**Convexity of the Objective Function:** For each task $i$, the term $\frac{\omega_i}{g_i^\top d}$ is convex in $d$ over the region where $g_i^\top d > 0$, as the function $\frac{1}{x}$ is convex for $x > 0$ and $\omega_i > 0$.

**Convexity of the Constraints:** The constraint $g_i^\top d > 0$ is linear in $d$, and the norm constraint $\|d\| \leq \epsilon$ is also convex.

Since Eq. 4 is a convex optimization problem, we define the Lagrangian for this problem as follows:

$$L(d, \lambda, \{\mu_i\}) = \frac{1}{k} \sum_{i=1}^{k} \frac{\omega_i}{g_i^\top d} + \lambda(\|d\|^2 - \epsilon^2) - \sum_{i=1}^{k} \mu_i(g_i^\top d), \tag{6}$$

where $\lambda \geq 0$ and $\mu_i \geq 0$ are Lagrange multipliers associated with the constraints. The multiplier $\lambda$ enforces the norm constraint, and $\mu_i$ enforces the positivity of the inner products $g_i^\top d > 0$. The following Karush-Kuhn-Tucker (KKT) conditions provide the necessary conditions for optimality:

**1) Primal Feasibility:** $g_i^\top d > 0$, $\|d\| \leq \epsilon$, $\forall i$. This ensures that the inner products are positive and the norm of $d$ does not exceed $\epsilon$.

**2) Dual Feasibility:** $\mu_i \geq 0$, $\lambda \geq 0$. This condition guarantees that the multipliers are non-negative, maintaining the validity of the constraints.

**3) Complementary Slackness:** $\mu_i(-g_i^\top d) = 0$, $\lambda(\|d\|^2 - \epsilon^2) = 0$. Since $g_i^\top d > 0$, it follows that $\mu_i = 0$ for all $i$.

**4) Stationarity:** The gradient of the Lagrangian with respect to $d$ must be zero at the optimum: $\nabla_d L = -\frac{1}{k} \sum_{i=1}^{k} \frac{\omega_i}{(g_i^\top d)^2} g_i + 2\lambda d = 0$. This equation describes the balance between the gradient contributions from each task and the regularization term from the norm constraint.

Given that $\mu_i = 0$, the stationarity condition simplifies to:

$$\sum_{i=1}^{k} \frac{\omega_i}{(g_i^\top d)^2} g_i = 2k\lambda d. \tag{7}$$

Following previous works (Navon et al., 2022; Ban & Ji, 2024), we similarly assume that the gradients of tasks are linearly independent otherwise it would imply reaching a Pareto stationary point. Hence, $d$ can be represented as a linear combination of task gradients: $d = \sum_{i=1}^{k} \alpha_i g_i$. Ignoring the parameter $2k\lambda$ in Eq. 7,

---

**Algorithm 1** PIVRG for MTL

1: **Input:** Model parameters $\theta_0$; Initial $\boldsymbol{\Delta m} = \mathbf{0}^\top$; Learning rate $\{\eta_t\}$; Train set and Validation set $D_t, D_v$.
2: **for** $t = 1$ **to** $T - 1$ **do**
3:     Compute gradients $\boldsymbol{\mathcal{G}}(\theta_t) = [g_1(\theta_t), \cdots, g_k(\theta_t)]$ on $D_t$
4:     Obtain weights $\boldsymbol{\omega_t}$ by Eq. 5 based on $\boldsymbol{\Delta m}$
5:     Solve Eq. 8 to obtain $\boldsymbol{\alpha_t}$
6:     Compute $d_t = \boldsymbol{\mathcal{G}}(\theta_t)\boldsymbol{\alpha_t}$
7:     Update the parameters $\theta_{t+1} = \theta_t - \eta_t d_t$
8:     Evaluate and update $\boldsymbol{\Delta m}$ on $D_v$
9: **end for**

---

which can be adjusted by the step size $\eta_t$, we obtain $\alpha_i = \frac{\omega_i}{(g_i^\top d)^2}$, i.e., $(g_i^\top d)^2 = \frac{\omega_i}{\alpha_i}$.

Let $\boldsymbol{\mathcal{G}} = (g_1, g_2, \ldots, g_k) \in \mathbb{R}^{n \times k}$ denote the matrix of task gradients. Then, we can express this in matrix form as:

$$(\boldsymbol{\mathcal{G}}^\top \boldsymbol{\mathcal{G}} \boldsymbol{\alpha})^2 = \frac{\boldsymbol{\omega}}{\boldsymbol{\alpha}}, \tag{8}$$

where the square operation is element-wise. Following (Ban & Ji, 2024), we treat Eq. 8 as a simple constrained nonlinear least squares problem, which can be efficiently solved using the `scipy` library. Our complete algorithmic procedure is summarized in algorithm 1. Note that for certain benchmarks lacking a validation set, to ensure consistency with other methods on the dataset, we use $\boldsymbol{\Delta m}$ from the training set to obtain the $\boldsymbol{\omega}$.

### 3.4 THEORETICAL ANALYSIS

In this section, we present a theoretical analysis of our method about its convergence to a Pareto stationary point, where a convex combination of task gradients becomes zero. As previously noted, we assume that task gradients remain linearly independent until the system reaches a Pareto stationary point. Formally, we adopt the following assumption, similarly used by Navon et al. (2022) and Ban & Ji (2024).

**Assumption 1.** *For the output sequence $\{\theta_t\}_{t=1}^{\infty}$ produced by the proposed method, the gradients of the tasks $g_{1,t}, g_{2,t}, \cdots, g_{k,t}$ remain linearly independent as long as the system has not reached a Pareto stationary point.*

In practice, this assumption generally holds during the optimization process, as the number of tasks $k$ is often much smaller than the dimension $n$ of the shared parameters $\theta$. The following assumption imposes differentiability and Lipschitz continuity on the loss functions, as also adopted by previous works (Liu et al., 2021a; Navon et al., 2022; Ban & Ji, 2024).

**Assumption 2.** *For each task, the loss function $\ell_i(\theta)$ is differentiable and L-smooth such that $\|\nabla\ell_i(\theta_1) - \nabla\ell_i(\theta_2)\| \leq L\|\theta_1 - \theta_2\|$ for any $\theta_1$ and $\theta_2$.*

Then, we can obtain the following convergence theorem:

**Theorem 1.** *Suppose Assumptions 1 and 2 hold. We set the stepsize $\eta_t = \frac{\sum_i \sqrt{\omega_{i,t}/\alpha_{i,t}}}{kL \sum_i \sqrt{\omega_{i,t}\alpha_{i,t}}}$. Then, the sequence $\{\theta_t\}_{t=1}^{\infty}$ has a subsequence that converges to a Pareto stationary point $\theta^*$.*

The detailed proof can be found in Appendix A.3. Our main idea is to show that the smallest singular value of $\mathcal{G}^\top\mathcal{G}$ gradually approaches zero as the number of optimization steps $t$ increases, thereby leading to the eventual convergence to a Pareto stationary point, where the gradients become linearly dependent.

## 4 EXPERIMENTS

### 4.1 PROTOCOLS

We evaluate the proposed PIVRG on a variety of multi-task learning (MTL) problems under both supervised learning and reinforcement learning settings to demonstrate its effectiveness. For multi-task supervised learning, we validate on the Scene Understanding benchmarks NYUv2 (Silberman et al., 2012) and Cityscapes (Cordts et al., 2016), regression tasks from QM9 (Blum & Reymond, 2009), and image-level classification with the CelebA (Liu et al., 2015) dataset. For multi-task reinforcement learning, we conduct experiments on the MT10 environment from the Meta-World benchmark (Yu et al., 2020c). Additionally, the ablation study demonstrates performance-level information in PIVRG can be integrated into existing methods to significantly improve their performance. Note that for the QM9 and CelebA benchmarks, which already have predefined validation sets, we use $\Delta m$ from the validation set to update $w$. For the NYUv2 and Cityscapes benchmarks, which lack validation sets, we use $\Delta m$ from the training set to update $w$ to maintain consistency with other methods on the dataset. Moreover, we visualize the optimization process of PIVRG on a 2-task toy example (Liu et al., 2021a) in Fig.2 in the appendix.

**Baselines:** We compare our proposed PIVRG described in Section 3 with the following methods in our experiments: Single-task learning (STL), Linear Scalarization (LS), Scale-Invariant (SI), Dynamic Weight Average (DWA) (Liu et al., 2019), Uncertainty Weighting (UW) (Kendall et al., 2018a), Multi-Gradient Descent Algorithm (MGDA) (Sener & Koltun, 2018), Random Loss Weighting (RLW) (Lin et al., 2021a), PCGrad (Yu et al., 2020b), GradDrop (Chen et al., 2020), CAGrad (Liu et al., 2021a), IMTL-G (Liu et al., 2021b), Nash-MTL (Navon et al., 2022), FAMO (Liu et al., 2024) and FairGrad (Ban & Ji, 2024).

**Evaluation Metrics:** Given that MTL does not inherently have a single objective and that metrics can vary across tasks, we follow previous works and focus on two overall performance metrics: (1) $\Delta m$, the average per-task performance drop of method $m$ relative to the STL baseline, which has been early defined in Eq. 3. (2) Mean Rank (MR): The average rank of each method across tasks (lower is better). A method achieves the best MR of 1 if it ranks first in all tasks.

### 4.2 MULTI-TASK SUPERVISED LEARNING

**Scene Understanding.** Following previous works (Navon et al., 2022; Liu et al., 2024; Ban & Ji, 2024), we evaluate PIVRG on the NYUv2 and Cityscapes datasets. NYUv2 (Silberman et al., 2012) contains 1449 densely annotated indoor images, with three pixel-level tasks: 13-class semantic segmentation, depth estimation, and surface normal prediction. Cityscapes (Cordts et al., 2016) is a similar dataset containing 5000 street-view images with two tasks: semantic segmentation and depth estimation. These scenarios test the effectiveness of MTL in complex, pixel-level predictions. We follow the setup in (Navon et al., 2022; Liu et al., 2024) using MTAN (Liu et al., 2021b), which adds task-specific attention modules on top of SegNet (Badrinarayanan et al., 2017). To align with previous works, the model is trained for 200 epochs with a learning rate of $10^{-4}$ for the first 100 epochs, decaying by half for the remaining epochs.

The results in Table 1 and Table 2 demonstrate the remarkable performance of our method. On the NYUv2 dataset, previous methods typically outperform the STL baseline on segmentation and

Table 1: Results on NYU-v2 (3-task) dataset. Each experiment is repeated 3 times with different random seeds and the average is reported. The detailed standard error is reported in the appendix.

| METHOD | SEGMENTATION | | DEPTH | | SURFACE NORMAL | | | | | MR ↓ | Δm(%) ↓ |
|---|---|---|---|---|---|---|---|---|---|---|---|
| | | | | | ANGLE DISTANCE ↓ | | WITHIN $t°$ ↑ | | | | |
| | mIoU ↑ | Pix Acc ↑ | Abs Err ↓ | Rel Err ↓ | Mean | Median | 11.25 | 22.5 | 30 | | |
| STL | 38.30 | 63.76 | 0.6754 | 0.2780 | 25.01 | 19.21 | 30.14 | 57.20 | 69.15 | | |
| LS | 39.29 | 65.33 | 0.5493 | 0.2263 | 28.15 | 23.96 | 22.09 | 47.50 | 61.08 | 11.78 | 5.59 |
| SI | 38.45 | 64.27 | 0.5354 | 0.2201 | 27.60 | 23.37 | 22.53 | 48.57 | 62.32 | 10.22 | 4.39 |
| RLW | 37.17 | 63.77 | 0.5759 | 0.2410 | 28.27 | 24.18 | 22.26 | 47.05 | 60.62 | 14.22 | 7.78 |
| DWA | 39.11 | 65.31 | 0.5510 | 0.2285 | 27.61 | 23.18 | 24.17 | 50.18 | 62.39 | 10.67 | 3.57 |
| UW | 36.87 | 63.17 | 0.5446 | 0.2260 | 27.04 | 22.61 | 23.54 | 49.05 | 63.65 | 10.33 | 4.05 |
| MGDA | 30.47 | 59.90 | 0.6070 | 0.2555 | 24.88 | 19.45 | 29.18 | 56.88 | 69.36 | 8.11 | 1.38 |
| PCGrad | 38.06 | 64.64 | 0.5550 | 0.2325 | 27.41 | 22.80 | 23.86 | 49.83 | 63.14 | 10.89 | 3.97 |
| GradDrop | 39.39 | 65.12 | 0.5455 | 0.2279 | 27.48 | 22.96 | 23.38 | 49.44 | 62.87 | 9.89 | 3.58 |
| CAGrad | 39.79 | 65.49 | 0.5486 | 0.2250 | 26.31 | 21.58 | 25.61 | 52.36 | 65.58 | 6.89 | 0.20 |
| IMTL-G | 39.35 | 65.60 | 0.5426 | 0.2256 | 26.02 | 21.19 | 26.20 | 53.13 | 66.24 | 6.11 | -0.76 |
| MoCo | **40.30** | **66.07** | 0.5575 | **0.2135** | 26.67 | 21.83 | 25.61 | 51.78 | 64.85 | 6.22 | 0.16 |
| Nash-MTL | 40.13 | 65.93 | **0.5261** | 0.2171 | 25.26 | 20.08 | 28.40 | 55.47 | 68.15 | 3.67 | -4.04 |
| FAMO | 38.88 | 64.90 | 0.5474 | 0.2194 | 25.06 | 19.57 | 29.21 | 56.61 | 68.98 | 5.33 | -4.10 |
| FairGrad | 39.74 | 66.01 | 0.5377 | 0.2236 | 24.84 | 19.60 | 29.26 | 56.58 | 69.16 | 3.44 | -4.66 |
| PIVRG | 39.90 | 65.74 | 0.5365 | 0.2243 | **24.30** | **18.80** | **30.95** | **58.26** | **70.38** | **2.33** | **-6.50** |

Table 2: Results on Cityscapes (2-task) and CelebA (40-task) datasets. Each experiment is repeated 3 times with different random seeds and the average is reported. The detailed standard error is reported in the appendix.

| METHOD | CITYSCAPES | | | | | | CELEBA | |
|---|---|---|---|---|---|---|---|---|
| | SEGMENTATION | | DEPTH | | MR ↓ | Δm(%) ↓ | MR ↓ | Δm(%) ↓ |
| | mIoU ↑ | Pix Acc ↑ | Abs Err ↓ | Rel Err ↓ | | | | |
| STL | 74.01 | 93.16 | 0.0125 | 27.77 | | | | |
| LS | 75.18 | 93.49 | 0.0155 | 46.77 | 8.75 | 22.60 | 7.65 | 4.15 |
| SI | 70.95 | 91.73 | 0.0161 | 33.83 | 11.25 | 14.11 | 9.43 | 7.20 |
| RLW | 74.57 | 93.41 | 0.0158 | 47.79 | 11.25 | 24.38 | 6.65 | 1.46 |
| DWA | 75.24 | 93.52 | 0.0160 | 44.37 | 8.50 | 21.45 | 8.32 | 3.20 |
| UW | 72.02 | 92.85 | 0.0140 | 30.13 | 7.75 | 5.89 | 6.95 | 3.23 |
| MGDA | 68.84 | 91.54 | 0.0309 | 33.50 | 11.75 | 44.14 | 12.88 | 14.85 |
| PCGrad | 75.13 | 93.48 | 0.0154 | 42.07 | 9.00 | 18.29 | 8.03 | 3.17 |
| GradDrop | 75.27 | 93.53 | 0.0157 | 47.54 | 8.00 | 23.73 | 9.45 | 3.29 |
| CAGrad | 75.16 | 93.48 | 0.0141 | 37.60 | 7.75 | 11.64 | 7.62 | 2.48 |
| IMTL-G | 75.33 | 93.49 | 0.0135 | 38.41 | 6.00 | 11.10 | 5.88 | 0.84 |
| Nash-MTL | 75.41 | 93.66 | 0.0129 | 35.02 | 3.50 | 6.82 | 6.30 | 2.84 |
| FAMO | 74.54 | 93.29 | 0.0145 | 32.59 | 8.25 | 8.13 | 5.97 | 1.21 |
| FairGrad | 75.72 | **93.68** | 0.0134 | 32.25 | 2.25 | 5.18 | 6.62 | 0.37 |
| PIVRG | **75.82** | 93.65 | **0.0126** | 27.87 | 1.50 | -0.54 | 3.25 | -0.96 |

depth estimation tasks but fail to surpass STL on the surface normal prediction task, indicating a task imbalance. In contrast, our method is the only one that consistently outperforms STL across all 3 tasks and 9 evaluation metrics, and achieves an impressive average rank of **2.33** and the best performance drop of **-6.50%**.

In the Cityscapes dataset, prior methods often exhibit better optimization on the segmentation task while underperforming on the depth estimation task. In contrast, PIVRG achieves more balanced results and is the first method to achieve a negative $\Delta m$ on this benchmark, which means that for the first time, an MTL method has surpassed the STL baseline in terms of average performance. This further highlights both the potential of MTL and the superiority of PIVRG. In Appendix B.3, we also show that our method not only achieves SOTA performance on the NYUv2 and Cityscapes benchmarks but also produces the lowest performance variance, indicating a fairer optimization.

**Image-Level Classification.** CelebA (Liu et al., 2015) is a large-scale facial attributes dataset containing over 200K images, annotated with 40 attributes such as smiling, wavy hair, and mustache. This scenario represents a 40-task MTL classification problem, where each task predicts a binary attribute. We follow the setup in (Liu et al., 2024) and use a 9-layer convolutional neural network (CNN) as the backbone, with task-specific linear layers. The method is trained for 15 epochs using the Adam optimizer with a learning rate of $3 \times 10^{-4}$ and a batch size of 256. The results are shown in Table 2. On this benchmark with as many as 40 tasks, PIVRG also shows state-of-the-art performance, achieving a negative $\Delta m$ for the first time, validating the superiority of our approach.

Table 3: Results on QM9 (11-task) dataset. Each experiment is repeated 3 times with different random seeds and the average is reported. The detailed standard error is reported in the appendix.

| Method | $\mu$ | $\alpha$ | $\epsilon_{HOMO}$ | $\epsilon_{LUMO}$ | $\langle R^2 \rangle$ | ZPVE | $U_0$ | $U$ | $H$ | $G$ | $c_v$ | MR↓ | $\Delta m(\%)$ ↓ |
|---|---|---|---|---|---|---|---|---|---|---|---|---|---|
| | | | | | | MAE↓ | | | | | | | |
| STL | 0.067 | 0.181 | 60.57 | 53.91 | 0.502 | 4.53 | 58.8 | 64.2 | 63.8 | 66.2 | 0.072 | | |
| LS | 0.106 | 0.325 | **73.57** | 89.67 | 5.19 | 14.06 | 143.4 | 144.2 | 144.6 | 140.3 | 0.128 | 9.09 | 177.6 |
| SI | 0.309 | 0.345 | 149.8 | 135.7 | **1.00** | 4.50 | 55.3 | **55.75** | **55.82** | **55.27** | 0.112 | 5.55 | 77.8 |
| RLW | 0.113 | 0.340 | 76.95 | 92.76 | 5.86 | 15.46 | 156.3 | 157.1 | 157.6 | 153.0 | 0.137 | 10.64 | 203.8 |
| DWA | 0.107 | 0.325 | 74.06 | 90.61 | 5.09 | 13.99 | 142.3 | 143.0 | 143.4 | 139.3 | 0.125 | 8.82 | 175.3 |
| UW | 0.386 | 0.425 | 166.2 | 155.8 | 1.06 | 4.99 | 66.4 | 66.78 | 66.80 | 66.24 | 0.122 | 7.27 | 108.0 |
| MGDA | 0.217 | 0.368 | 126.8 | 104.6 | 3.22 | 5.69 | 88.37 | 89.4 | 89.32 | 88.01 | 0.120 | 8.91 | 120.5 |
| PCGRAD | 0.106 | 0.293 | 75.85 | 88.33 | 3.94 | 9.15 | 116.36 | 116.8 | 117.2 | 114.5 | 0.110 | 7.27 | 125.7 |
| CAGRAD | 0.118 | 0.321 | 83.51 | 94.81 | 3.21 | 6.93 | 113.99 | 114.3 | 114.5 | 112.3 | 0.116 | 8.18 | 112.8 |
| IMTL-G | 0.136 | 0.287 | 98.31 | 93.96 | 1.75 | 5.69 | 101.4 | 102.4 | 102.0 | 100.1 | 0.096 | 7.18 | 77.2 |
| NASH-MTL | **0.102** | 0.248 | 82.95 | **81.89** | 2.42 | 5.38 | 74.5 | 75.02 | 75.10 | 74.16 | 0.093 | 4.36 | 62.0 |
| FAMO | 0.15 | 0.30 | 94.0 | 95.2 | 1.63 | 4.95 | 70.82 | 71.2 | 71.2 | 70.3 | 0.10 | 5.73 | 58.5 |
| FAIRGRAD | 0.117 | 0.253 | 87.57 | 84.00 | 2.15 | 5.07 | 70.89 | 71.17 | 71.21 | 70.88 | 0.095 | 4.73 | 57.9 |
| PIVRG | 0.125 | **0.226** | 94.80 | 81.98 | 1.41 | **3.87** | 57.79 | 57.90 | 58.09 | 57.86 | **0.085** | **3.00** | **33.6** |

**Multi-Task Regression.** QM9 (Blum & Reymond, 2009) is a commonly used benchmark in graph neural networks, containing over 130K organic molecules represented as graphs. Each task predicts one of 11 molecular properties, which vary in scale. This setting evaluates the ability of MTL methods to balance task variations. Predicting molecular properties in the QM9 dataset presents a major challenge for MTL methods due to the large number of tasks and the substantial variation in loss scales. In our experiments, we train each method for 300 epochs and employ a learning rate scheduler to adjust the learning rate, consistent with prior works.

The results are presented in Figure 1 and Table 3. PIVRG achieves the best performance in terms of both MR and $\Delta m$. On the QM9 benchmark, where task difficulty is highly imbalanced, prior methods have struggled to optimize all tasks effectively, leading to a large overall $\Delta m$. By incorporating performance-level information and employing dynamic weight allocation to control variance, PIVRG reduces the average $\Delta m$ by over 20%. Meanwhile, the results in Figure 1 also show that PIVRG achieves the smallest performance variance while obtaining the optimal $\Delta m$, further validating the effectiveness of the performance-informed dynamic weight allocation strategy. This also underscores the potential of MTL approaches and the distinct advantages of PIVRG in addressing task imbalance and achieving superior optimization across tasks.

## 4.3 Multi-Task Reinforcement Learning

We further evaluate our method on the MT10 benchmark, which includes 10 robotic manipulation tasks from the MetaWorld environment (Yu et al., 2020c), where the objective is to learn a single policy that generalizes across various tasks such as pick and place, and opening doors. We follow the methodologies outlined in (Navon et al., 2022; Liu et al., 2024) and adopt Soft Actor-Critic (SAC) (Haarnoja et al., 2018) as the underlying algorithm. Our implementation utilizes the MTRL codebase used in (Navon et al., 2022; Ban & Ji, 2024) and trains the model for 2 million steps with a batch size of 1280. We compare our proposed PIVRG with Multi-task SAC (MTL SAC) (Yu et al., 2020c), Multi-task SAC with task encoder (MTL SAC + TE) (Yu et al., 2020c), Multi-headed SAC (MH SAC) (Yu et al., 2020c), PCGrad (Yu et al., 2020b), CAGrad (Liu et al., 2021a), MoCo (Fernando et al., 2023), Nash-MTL (Navon et al., 2022), FAMO (Liu et al., 2024) and FairGrad (Ban & Ji, 2024).

Table 4: Results on MT10 benchmark. Average over 10 random seeds.

| Method | Success Rate (mean ± stderr) |
|---|---|
| STL | 0.90 ± 0.03 |
| MTL SAC | 0.49 ± 0.07 |
| MTL SAC + TE | 0.54 ± 0.05 |
| MH SAC | 0.61 ± 0.04 |
| PCGRAD | 0.72 ± 0.02 |
| CAGRAD | 0.83 ± 0.05 |
| MOCO | 0.75 ± 0.05 |
| NASH-MTL | 0.91 ± 0.03 |
| FAMO | 0.83 ± 0.05 |
| FAIRGRAD | 0.84 ± 0.07 |
| PIVRG | **0.96** ± 0.02 |

The results are shown in Table 4. Each method is evaluated every 10,000 steps, and the best average success rate over 10 random seeds throughout the entire training period is reported. In this context, we directly utilize the success rate to update $\omega$. The results indicate that PIVRG achieves state-of-the-art performance on the MT10 benchmark, with an access rate approaching 100%.

Table 5: Results of integrating our performance-informed weighting strategy into existing methods on the NYU-v2 (3-task) dataset. Each experiment is repeated 3 times with different random seeds and the average is reported.

| Method | Segmentation | | Depth | | Surface Normal | | | | | $\Delta m\%\downarrow$ | $\text{Var}[\Delta m_i]\downarrow$ |
| | mIoU ↑ | Pix Acc ↑ | Abs Err ↓ | Rel Err ↓ | Angle Distance ↓ | | Within $t°$ ↑ | | | | |
| | | | | | Mean | Median | 11.25 | 22.5 | 30 | | |
| LS | 39.29 | 65.33 | 0.5493 | 0.2263 | 28.15 | 23.96 | 22.09 | 47.50 | 61.08 | 5.59 | 259.1 |
| PI-LS | 40.59 | 66.24 | 0.5330 | 0.2191 | 26.66 | 21.80 | 25.19 | 51.94 | 65.05 | **-0.06** | **173.2** |
| RLW | 37.17 | 63.77 | 0.5759 | 0.2410 | 28.27 | 24.18 | 22.26 | 47.05 | 60.62 | 7.78 | 205.3 |
| PI-RLW | 39.86 | 64.86 | 0.5744 | 0.2410 | 27.38 | 22.84 | 22.75 | 49.58 | 63.26 | **4.52** | **170.5** |
| DWA | 39.11 | 65.31 | 0.5510 | 0.2285 | 27.61 | 23.18 | 24.17 | 50.18 | 62.39 | 3.57 | 191.9 |
| PI-DWA | 40.55 | 66.31 | 0.5480 | 0.2261 | 26.63 | 21.97 | 25.03 | 51.42 | 64.67 | **0.78** | **158.8** |
| UW | 36.87 | 63.17 | 0.5446 | 0.2260 | 27.04 | 22.61 | 23.54 | 49.05 | 63.65 | 4.05 | 190.7 |
| PI-UW | 40.23 | 65.84 | 0.5182 | 0.2147 | 26.13 | 21.14 | 26.25 | 53.09 | 66.09 | **-1.71** | **158.7** |
| MGDA | 30.47 | 59.90 | 0.6070 | 0.2555 | 24.88 | 19.45 | 29.18 | 56.88 | 69.36 | 1.38 | 68.6 |
| PI-MGDA | 35.45 | 63.04 | 0.6025 | 0.2364 | 24.32 | 18.59 | 31.06 | 58.73 | 70.62 | **-3.45** | **36.6** |
| Nash-MTL | 40.13 | 65.93 | 0.5261 | 0.2171 | 25.26 | 20.08 | 28.40 | 55.47 | 68.15 | -4.04 | 108.0 |
| PI-Nash-MTL | 42.14 | 66.83 | 0.5317 | 0.2259 | 24.79 | 19.46 | 29.46 | 56.93 | 69.30 | **-5.77** | **70.7** |

## 4.4 Integrating Performance-Informed Weighting into Existing Methods

Previous loss-based and gradient-based methods have often overlooked performance-level information, leading to a lack of clarity regarding task difficulty during the training process. We propose to integrate our performance-informed weighting strategy into these methods to enhance fairness in optimization. Specifically, for loss-based approaches, we adjust the initial loss $\mathcal{L} = (\ell_1, \ell_2, \cdots, \ell_k)$ using weights $\boldsymbol{\omega}$ to reflect the current optimization progress of different tasks, replacing $\mathcal{L}$ with $\mathcal{L}' = \boldsymbol{\omega} \odot \mathcal{L}$.

For gradient-based methods, since the motivation behind the aggregation algorithms varies, it is necessary to analyze each method individually to incorporate $\boldsymbol{\omega}$ into the design of the aggregation process. For instance, Nash-MTL maximizes the sum of log utilities, we thus replace the original equal summation $(1, 1, \cdots, 1)$ with a weighted sum $(\omega_1, \omega_2, \cdots, \omega_k)$.

We apply the performance-informed weighting strategy to a series of MTL methods, including LS, RLW (Lin et al., 2021a), DWA (Liu et al., 2019), UW (Kendall et al., 2018a), MGDA (Sener & Koltun, 2018), and Nash-MTL (Navon et al., 2022), and evaluate their performance on the NYUv2 benchmark. Table 5 shows that incorporating performance-level information and integrating dynamic weighting can bring significant performance improvements for these methods. $\text{Var}[\Delta m_i]$ is also reduced, which indicates a notable alleviation of task imbalance.

## 5 Conclusion, Limitations and Future Work

In this paper, we propose PIVRG, a novel performance-informed variance reduction gradient aggregation approach. Building on the observation that previous loss-based and gradient-based methods exhibit common task imbalance across standard benchmarks, we point out the necessity of incorporating performance-level information to better represent fairness across tasks during the optimization process. Specifically, we use performance variance across tasks as a fairness indicator and introduce a dynamic weighting strategy aimed at gradually reducing this variance. Extensive experiments show that PIVRG achieves state-of-the-art performance across various benchmarks. The experimental results also show that incorporating our dynamic weighting strategy into existing loss-based and gradient-based methods not only significantly improves overall performance but also reduces performance variance across tasks, leading to a more balanced optimization process.

**Limitations and Future Work.** In this work, we regard performance variance across tasks as a fairness indicator and design a dynamic weighting strategy to progressively reduce this variance. However, there are numerous ways to incorporate performance-level information, and we would like to explore more effective fairness indicators in our future work. Additionally, our underlying optimization objective is not fixed, and future work may explore alternative designs and approaches to further enhance fairness and efficiency in multi-task learning.

ETHICS STATEMENT

In this research, we are committed to exploring the fairness in multi-task learning, particularly through the lens of performance-informed variance reduction. The datasets used for our experiments, including NYUv2, Cityscapes, QM9, CelebA, and MT10, are publicly available and widely used within the research community. We ensure that our use of these datasets adheres to the respective licensing agreements and ethical guidelines established by the dataset creators. We acknowledge the potential implications of our findings on fairness in machine learning systems. Our proposed methods aim to reduce task imbalance and enhance performance equity across different tasks, thereby mitigating biases that may arise in multi-task learning frameworks. We are committed to transparency and responsible dissemination of our results, and we encourage further exploration of the ethical implications of our methodologies.

REPRODUCIBILITY STATEMENT

To ensure the reproducibility of our work, we have essentially adhered to the experimental setups of prior methods, as detailed in Section 4 where we report specific settings for each experiment. Additionally, all experimental results presented in the main text are averages obtained from multiple runs to mitigate the impact of randomness, with standard errors provided in Appendix B.2 for further clarity. The source code will be made publicly available soon, along with checkpoint files corresponding to each experiment. These checkpoint files may yield slightly improved results compared to those reported in the main text, as they represent the best outcomes from multiple runs.

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

## A    THEORETICAL ANALYSIS

### A.1    PROOF OF PROPOSITION 1.

**Proposition 1.** *Let $\Delta m^{max}$ be the maximum value of $\boldsymbol{\Delta m}$, and $\Delta m^{min}$ be the minimum value. Define $s = \min\left(\frac{1}{\underline{\omega}}, \overline{\omega}\right)$. Then, for $\tau > \frac{\Delta m^{max} - \Delta m^{min}}{\log s}$, Property 3 is satisfied.*

*Proof.* Define $\omega_{min}$ and $\omega_{max}$ as the minimum and maximum value of $\boldsymbol{\omega}$ respectively, since $\omega_i$ is positively correlated with $\Delta m_i$, we have

$$\omega_{max} = \frac{k \cdot \exp(\Delta m^{max}/\tau)}{\sum_{j=1}^{k} \exp(\Delta m_j/\tau)}, \quad \omega_{min} = \frac{k \cdot \exp(\Delta m^{min}/\tau)}{\sum_{j=1}^{k} \exp(\Delta m_j/\tau)}.$$

We notice that

$$\frac{\omega_{max}}{\omega_{min}} = \frac{\exp(\Delta m^{max}/\tau)}{\exp(\Delta m^{min}/\tau)} = \exp(\frac{\Delta m^{max} - \Delta m^{min}}{\tau}) \stackrel{\text{def}}{=} s.$$

By incorporating property 2, we have

$$k = \sum_{i=1}^{k} \omega_i \leq \sum_{i=1}^{k} \omega_{max} \leq \sum_{i=1}^{k} s\, \omega_{min} = ks\, \omega_{min},$$

showing that $\omega_{min} \geq \frac{1}{s}$. In the same way, we have $\omega_{max} \leq s$. Thus by setting $s = \min(\frac{1}{\underline{\omega}}, \overline{\omega})$, we can derive that

$$\omega_i \in [\frac{1}{s}, s] \subseteq [\underline{\omega}, \overline{\omega}].$$

Notice that $\frac{\Delta m^{max} - \Delta m^{min}}{\tau} = \log s$, then for $\tau > \frac{\Delta m^{max} - \Delta m^{min}}{\log s}$, Property 3 is satisfied. In practice, we pre-define a threshold $\tau^*$, and let $\tau = \max(\frac{\Delta m^{max} - \Delta m^{min}}{\log s}, \tau^*)$ to further guarantee the smoothness and contraints. $\qquad\square$

### A.2    PROOF OF PROPOSITION 2.

**Proposition 2.** *For $\boldsymbol{\omega}$ satisfying the three properties above, we have the following approximation:*

$$\text{Var}[\Delta m_i] \approx \frac{\tau^2}{k} \boldsymbol{\omega}^\top \boldsymbol{\omega} - \tau^2.$$

*Proof.* Following the notation in the main paper, let $\Delta m = \frac{1}{k}\sum_{i=1}^{k} \Delta m_i$ and $\text{Var}[\Delta m_i] = \sigma^2$. Define $\{\epsilon_i\}_{i=1}^{k}$ such that $\Delta m_i = \Delta m + \epsilon_i$, thus $\mathbb{E}[\epsilon_i] = 0$.

We know that $\omega_i = \frac{k \cdot \exp(\Delta m_i/\tau)}{\sum_{j=1}^{k} \exp(\Delta m_j/\tau)}$. For the numerator,

$$k \cdot \exp(\Delta m_i/\tau) = k \cdot \exp\left(\frac{\Delta m}{\tau} + \frac{\epsilon_i}{\tau}\right) = k \cdot \exp\left(\frac{\Delta m}{\tau}\right) \cdot \exp\left(\frac{\epsilon_i}{\tau}\right)$$

$$\approx k \cdot \exp\left(\frac{\Delta m}{\tau}\right)\left(1 + \frac{\epsilon_i}{\tau}\right),$$

since $\tau$ is generally large enough such that $\frac{\epsilon_i}{\tau}$ is pretty small. Similarly, for the denominator, we have

$$\sum_{j=1}^{k} \exp(\Delta m_j/\tau) \approx \sum_{j=1}^{k} \exp\left(\frac{\Delta m}{\tau}\right)\left(1 + \frac{\epsilon_j}{\tau}\right) = \exp\left(\frac{\Delta m}{\tau}\right)\left(k + \sum_{j=1}^{k} \frac{\epsilon_j}{\tau}\right)$$

$$\approx k \cdot \exp\left(\frac{\Delta m}{\tau}\right).$$

Thus, $\omega_i \approx \frac{k\left(1+\frac{\epsilon_i}{\tau}\right)}{k} = 1 + \frac{\epsilon_i}{\tau}$. Therefore, we can deduce

$$\mathbb{E}[\omega_i] \approx 1 + \frac{\mathbb{E}[\epsilon_i]}{\tau} = 1,$$

$$\mathbb{E}[\omega_i^2] \approx 1 + \frac{2\mathbb{E}[\epsilon_i]}{\tau} + \frac{\mathbb{E}[\epsilon_i^2]}{\tau^2} = 1 + \frac{\mathbb{E}[\epsilon_i^2]}{\tau^2}.$$

In fact, only $\mathbb{E}[\omega_i^2]$ is approximated, as $\boldsymbol{\omega}$ always satisfies $\mathbf{1}^\top \boldsymbol{\omega} = k$, which implies that $\mathbb{E}[\omega_i] = 1$. On the other hand,

$$\sigma^2 = \mathrm{Var}[\Delta m_i] = \mathbb{E}[\Delta m_i^2] - (\mathbb{E}[\Delta m_i])^2 = \frac{\sum_{i=1}^k (\Delta m + \epsilon_i)^2}{k} - \Delta m^2$$

$$= \frac{k\Delta m^2 + 2\Delta m \sum_{i=1}^k \epsilon_i + \sum_{i=1}^k \epsilon_i^2}{k} - \Delta m^2 = \frac{\sum_{i=1}^k \epsilon_i^2}{k},$$

which implies

$$\sum_{i=1}^k \epsilon_i^2 = k\sigma^2 \Rightarrow \mathbb{E}[\epsilon_i^2] = \sigma^2.$$

Then we can derive that

$$\mathrm{Var}[\omega_i] = \mathbb{E}[\omega_i^2] - (\mathbb{E}[\omega_i])^2 \approx 1 + \frac{\sigma^2}{\tau^2} - 1 = \frac{\sigma^2}{\tau^2} = \frac{\mathrm{Var}[\Delta m_i]}{\tau^2}.$$

From another perspective,

$$\mathrm{Var}[\omega_i] = \mathbb{E}[\omega_i^2] - (\mathbb{E}[\omega_i])^2 = \frac{\boldsymbol{\omega}^\top \boldsymbol{\omega}}{k} - 1,$$

which gives the final conclusion

$$\mathrm{Var}[\Delta m_i] \approx \frac{\tau^2}{k} \boldsymbol{\omega}^\top \boldsymbol{\omega} - \tau^2.$$

$\square$

### A.3 Proof of Theorem 1.

**Theorem 1.** *Suppose Assumptions 1 and 2 hold. We set the stepsize $\eta_t = \frac{\sum_i \sqrt{\omega_{i,t}/\alpha_{i,t}}}{kL \sum_i \sqrt{\omega_{i,t}\alpha_{i,t}}}$. Then, the sequence $\{\theta_t\}_{t=1}^\infty$ has a subsequence that converges to a Pareto stationary point $\theta^*$.*

*Proof.* Since $g_i^\top d = \sqrt{\frac{\omega_i}{\alpha_i}}$ and $d = \sum_{i=1}^k \alpha_i g_i$, we have $\|d\|^2 = \sum_i \alpha_i g_i^\top d = \sum_i \sqrt{\omega_i \alpha_i}$. Given that each loss function $\ell_i(\theta)$ is L-smooth, we have

$$\ell_i(\theta_{t+1}) \le \ell_i(\theta_t) - \eta_t g_{i,t}^\top d_t + \frac{L}{2}\|\eta_t d_t\|^2 = \ell_i(\theta_t) - \eta_t \sqrt{\frac{\omega_{i,t}}{\alpha_{i,t}}} + \frac{L}{2}\eta_t^2 \|d_t\|^2$$

$$= \ell_i(\theta_t) - \eta_t \sqrt{\frac{\omega_{i,t}}{\alpha_{i,t}}} + \frac{L\eta_t^2}{2}\left(\sum_{j=1}^k \sqrt{\omega_{j,t}\alpha_{j,t}}\right).$$

Set the learning rate $\eta_t = \frac{\sum_i \sqrt{\omega_{i,t}/\alpha_{i,t}}}{kL \sum_i \sqrt{\omega_{i,t}\alpha_{i,t}}}$. Consider the averaged loss function $\mathcal{L}(\theta) = \frac{1}{k}\sum_i \ell_i(\theta)$, we have

$$\mathcal{L}(\theta_{t+1}) \le \mathcal{L}(\theta_t) - \eta_t \frac{1}{k}\sum_{i=1}^k \sqrt{\frac{\omega_{i,t}}{\alpha_{i,t}}} + \frac{L\eta_t^2}{2}\left(\sum_{i=1}^k \sqrt{\omega_{i,t}\alpha_{i,t}}\right)$$

$$= \mathcal{L}(\theta_t) - L\eta_t^2\left(\sum_{i=1}^k \sqrt{\omega_{i,t}\alpha_{i,t}}\right) + \frac{L\eta_t^2}{2}\left(\sum_{i=1}^k \sqrt{\omega_{i,t}\alpha_{i,t}}\right)$$

$$= \mathcal{L}(\theta_t) - \frac{L\eta_t^2}{2}\left(\sum_{i=1}^k \sqrt{\omega_{i,t}\alpha_{i,t}}\right).$$

We can observe that $\sum_{r=0}^{t} \frac{L\eta_r^2}{2}(\sum_{i=1}^{k}\sqrt{\omega_{i,r}\alpha_{i,r}}) \leq \mathcal{L}(\theta_0) - \mathcal{L}(\theta_{t+1})$. Then, we get

$$\sum_{r=0}^{\infty} \frac{L\eta_r^2}{2}(\sum_{i=1}^{k}\sqrt{\omega_{i,r}\alpha_{i,r}}) = \frac{1}{2Lk^2}\sum_{r=0}^{\infty}\frac{\sum_{i=1}^{k}(\sqrt{\omega_{i,r}/\alpha_{i,r}})^2}{\sum_{i=1}^{k}\sqrt{\omega_{i,r}\alpha_{i,r}}} < \infty.$$

Then, it can be obtained that

$$\lim_{r\to\infty} \frac{\sum_{i=1}^{k}(\sqrt{\omega_{i,r}/\alpha_{i,r}})^2}{\sum_{i=1}^{k}\sqrt{\omega_{i,r}\alpha_{i,r}}} = 0. \tag{9}$$

From Eq. 8, we get

$$\left\|\sqrt{\frac{\boldsymbol{\omega_t}}{\boldsymbol{\alpha_t}}}\right\| \geq \sigma_k(\boldsymbol{\mathcal{G}}_t^\top \boldsymbol{\mathcal{G}}_t)\|\boldsymbol{\alpha_t}\|,$$

where $\sigma_k(\boldsymbol{\mathcal{G}}_t^\top \boldsymbol{\mathcal{G}}_t)$ is the smallest singular value of matrix $\boldsymbol{\mathcal{G}}_t^\top \boldsymbol{\mathcal{G}}_t$. Denote $\mathbf{1} = [1, \cdots, 1]^\top$ as the length-$k$ vector whose elements are all 1. Note that we have

$$\left\|\sqrt{\frac{\boldsymbol{\omega}}{\boldsymbol{\alpha}}}\right\|^2 = \sum_{i=1}^{k}\frac{\omega_i}{\alpha_i} \leq (\sum_{i=1}^{k}\sqrt{\frac{\omega_i}{\alpha_i}})^2 = \left\|\sqrt{\frac{\boldsymbol{\omega}}{\boldsymbol{\alpha}}}\right\|_1^2,$$

$$\|\boldsymbol{\alpha}\|_1 = \mathbf{1}^\top\boldsymbol{\alpha} \leq \|\mathbf{1}\|\cdot\|\boldsymbol{\alpha}\| = \sqrt{k}\|\boldsymbol{\alpha}\|.$$

Combine the above inequalities, we get

$$\left\|\sqrt{\frac{\boldsymbol{\omega_t}}{\boldsymbol{\alpha_t}}}\right\|_1 \geq \left\|\sqrt{\frac{\boldsymbol{\omega_t}}{\boldsymbol{\alpha_t}}}\right\| \geq \sigma_k(\boldsymbol{\mathcal{G}}_t^\top \boldsymbol{\mathcal{G}}_t)\|\boldsymbol{\alpha_t}\| \geq \frac{1}{\sqrt{k}}\sigma_k(\boldsymbol{\mathcal{G}}_t^\top \boldsymbol{\mathcal{G}}_t)\|\boldsymbol{\alpha_t}\|_1.$$

Then, we have

$$\frac{\sum_{i=1}^{k}\sqrt{\frac{\omega_{i,t}}{\alpha_{i,t}}}}{\sum_{i=1}^{k}\alpha_{i,t}} \geq \frac{1}{\sqrt{k}}\sigma_k(\boldsymbol{\mathcal{G}}_t^\top \boldsymbol{\mathcal{G}}_t). \tag{10}$$

Furthermore,

$$\frac{\sum_{i=1}^{k}\sqrt{\frac{\omega_{i,t}}{\alpha_{i,t}}}}{\sum_{i=1}^{k}\alpha_{i,t}} = \frac{(\sum_{i=1}^{k}\sqrt{\frac{\omega_{i,t}}{\alpha_{i,t}}})^2}{(\sum_{i=1}^{k}\alpha_{i,t})\cdot(\sum_{i=1}^{k}\sqrt{\frac{\omega_{i,t}}{\alpha_{i,t}}})} = \frac{(\sum_{i=1}^{k}\sqrt{\frac{\omega_{i,t}}{\alpha_{i,t}}})^2}{\sum_{i=1}^{k}\sqrt{\omega_{i,t}\alpha_{i,t}} + \sum_{i=1}^{k}\sum_{j=1,j\neq i}^{k}\alpha_{i,t}\sqrt{\frac{\omega_{j,t}}{\alpha_{j,t}}}}$$

$$\leq \frac{(\sum_{i=1}^{k}\sqrt{\frac{\omega_{i,t}}{\alpha_{i,t}}})^2}{\sum_{i=1}^{k}\sqrt{\omega_{i,t}\alpha_{i,t}}}. \tag{11}$$

For any fixed $k$, it can be concluded from Eq. 9, Eq. 10, and Eq. 11 that

$$\lim_{t\to\infty}\sigma_k(\boldsymbol{\mathcal{G}}_t^\top \boldsymbol{\mathcal{G}}_t) = 0.$$

Since the sequence $\mathcal{L}(\theta_t)$ is monotonically decreasing, we know the sequence $\theta_t$ is in the compact sublevel set $\{\theta|\mathcal{L}(\theta) \leq \mathcal{L}(\theta_0)\}$. Then, there exists a subsequence $\theta_{t_j}$ that converges to $\theta^\star$ where we have $\sigma_k(\boldsymbol{\mathcal{G}}_\star^\top \boldsymbol{\mathcal{G}}_\star) = 0$ and $\boldsymbol{\mathcal{G}}_\star$ denotes the matrix of multiple gradients at $\theta^\star$. Therefore, the gradients at $\theta^\star$ are linearly dependent, and $\theta^\star$ is Pareto stationary.

$\square$

## B EXPERIMENTS DETAILS

### B.1 TOY EXAMPLE

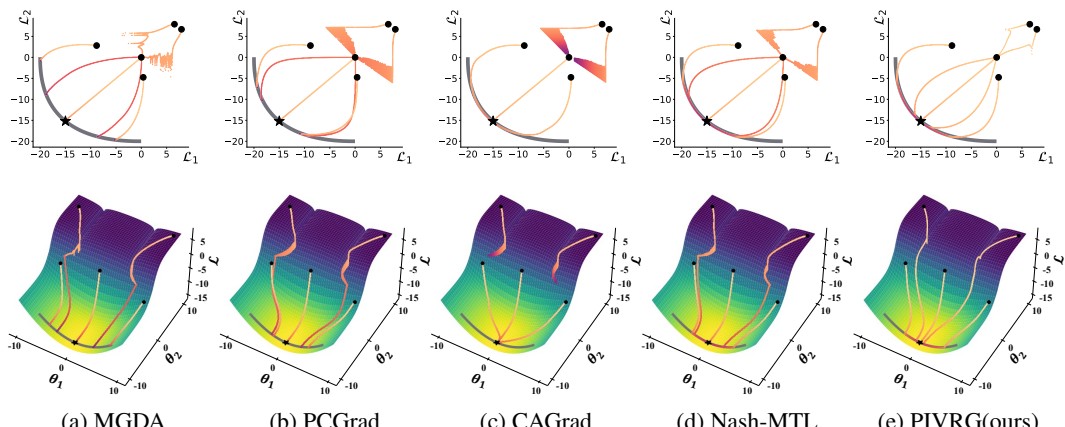

(a) MGDA     (b) PCGrad     (c) CAGrad     (d) Nash-MTL     (e) PIVRG(ours)

Figure 2: Comparison of MTL approaches on a challenging synthetic two-task benchmark (Liu et al., 2021a; Navon et al., 2022). We visualize optimization trajectories w.r.t. objectives value ($\mathcal{L}_1$ and $\mathcal{L}_2$, top row), and cumulative objective w.r.t. parameters ($\theta_1$ and $\theta_2$, bottom row). The starting points are indicated by black dots (●), and the Pareto front (see Definition 1) is represented by thick gray lines (▬).

Following Navon et al. (2022); Senushkin et al. (2023), we employ a two-task toy example presented in (Liu et al., 2021a). The two tasks $\mathcal{L}_1(\theta)$ and $\mathcal{L}_2(\theta)$ are defined on $\theta = (\theta_1, \theta_2)^\top \in \mathbb{R}^2$,

$$\mathcal{L}_1(\theta) = f_1(\theta)g_1(\theta) + f_2(\theta)h_1(\theta)$$
$$\mathcal{L}_2(\theta) = f_1(\theta)g_2(\theta) + f_2(\theta)h_2(\theta),$$

where the functions are defined as follows:

$$f_1(\theta) = \max\big(\tanh(0.5\theta_2), 0\big)$$
$$f_2(\theta) = \max\big(\tanh(-0.5\theta_2), 0\big)$$
$$g_1(\theta) = \log\Big(\max\big(|0.5(-\theta_1 - 7) - \tanh(-\theta_2)|, 0.000005\big)\Big) + 6$$
$$g_2(\theta) = \log\Big(\max\big(|0.5(-\theta_1 + 3) - \tanh(-\theta_2) + 2|, 0.000005\big)\Big) + 6$$
$$h_1(\theta) = \big((-\theta_1 + 7)^2 + 0.1(-\theta_2 - 8)^2\big)/10 - 20$$
$$h_2(\theta) = \big((-\theta_1 - 7)^2 + 0.1(-\theta_2 - 8)^2\big)/10 - 20.$$

Following (Navon et al., 2022; Liu et al., 2024; Ban & Ji, 2024), we use five distinct starting points $\{(-8.5, 7.5), (0, 0), (9.0, 9.0), (-7.5, -0.5), (9.0, -1.0)\}$. The Adam optimizer is employed with a learning rate of $1 \times 10^{-3}$. The 2D and 3D optimization trajectories are shown in 2. On one hand, while other MTL methods (Fig.1a to 1d) exhibit oscillations around local minima, leading to noisy optimization trajectories, our approach can swiftly escape these regions of local minima through guidance from the performance-informed weighting strategy. On the other hand, approaches designed to find a Pareto-stationary solution halt upon reaching the Pareto front (e.g. Fig.1a and Fig.1b), but PIVRG continues to transfer along the Pareto front and converges to a more balanced Pareto-optimal solution.

### B.2 EXPERIMENTAL RESULTS WITH STANDARD ERRORS

We followed the experimental setup from Navon et al. (2022); Liu et al. (2024); Ban & Ji (2024), and the results for the baseline methods are taken from their original papers. Below, we present PIVRG's results along with standard errors.

Table 6: Results on NYU-v2 dataset (3 tasks). Each experiment is repeated over 3 random seeds and the mean and stderr are reported.

| Method | Segmentation | | Depth | | Surface Normal | | | | | $\Delta m(\%) \downarrow$ |
|---|---|---|---|---|---|---|---|---|---|---|
| | | | | | Angle Dist $\downarrow$ | | Within $t^\circ \uparrow$ | | | |
| | mIoU $\uparrow$ | Pix Acc $\uparrow$ | Abs Err $\downarrow$ | Rel Err $\downarrow$ | Mean | Median | 11.25 | 22.5 | 30 | |
| PIVRG (mean) | 39.90 | 65.74 | 0.5365 | 0.2243 | 24.30 | 18.80 | 30.95 | 58.26 | 70.38 | -6.50 |
| PIVRG (stderr) | ±0.43 | ±0.21 | ±0.0007 | ±0.0014 | ±0.07 | ±0.09 | ±0.12 | ±0.18 | ±0.16 | ±0.24 |

Table 7: Results on QM-9 dataset (11 tasks). Each experiment is repeated over 3 random seeds and the mean and stderr are reported.

| Method | $\mu$ | $\alpha$ | $\epsilon_{\text{HOMO}}$ | $\epsilon_{\text{LUMO}}$ | $\langle R^2 \rangle$ | ZPVE | $U_0$ | $U$ | $H$ | $G$ | $c_v$ | $\Delta m(\%) \downarrow$ |
|---|---|---|---|---|---|---|---|---|---|---|---|---|
| | | | | | | MAE $\downarrow$ | | | | | | |
| PIVRG (mean) | 0.125 | 0.226 | 94.80 | 81.98 | 1.41 | 3.87 | 57.79 | 57.90 | 58.09 | 57.86 | 0.085 | 33.6 |
| PIVRG (stderr) | ±0.0022 | ±0.0078 | ±2.829 | ±1.349 | ±0.0301 | ±0.0438 | ±0.68 | ±0.72 | ±0.70 | ±0.68 | ±0.0005 | ±2.31 |

Table 8: Results on CityScapes (2 tasks) and CelebA (40 tasks) datasets. Each experiment is repeated over 3 random seeds and the mean and stderr are reported.

| Method | CityScapes | | | | | CelebA |
|---|---|---|---|---|---|---|
| | Segmentation | | Depth | | $\Delta m(\%) \downarrow$ | $\Delta m(\%) \downarrow$ |
| | mIoU $\uparrow$ | Pix Acc $\uparrow$ | Abs Err $\downarrow$ | Rel Err $\downarrow$ | | |
| PIVRG (mean) | 75.82 | 93.65 | 0.0126 | 27.87 | -0.54 | -0.96 |
| PIVRG (stderr) | ±0.05 | ±0.04 | ±0.0002 | ±0.24 | ±0.34 | ±0.34 |

## B.3 ADDITIONAL RESULTS ON PERFORMANCE VARIANCE

In Fig. 3 and Fig. 4, we show that both $\boldsymbol{\omega}^\top \boldsymbol{\omega}$ and $\text{Var}[\Delta m_i]$ decrease progressively throughout the optimization process, validating the effectiveness of our dynamic weights which serve as regularizers. In Table 9, 10 and 11, we compare the detailed performance drop $\Delta m$ and performance variance $\text{Var}[\Delta m_i]$ with existing methods, the results show that PIVRG not only achieves SOTA performance on various benchmarks but also produces the lowest performance variance, indicating a fairer optimization.

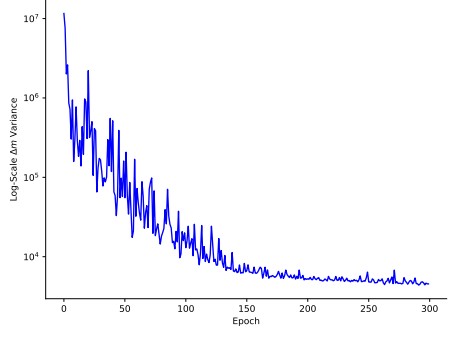 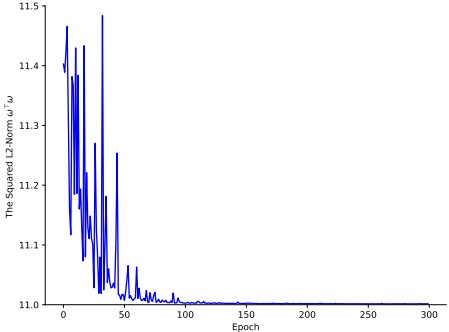

Figure 3: Performance Variance on QM9 dataset.  Figure 4: The squared $L^2$ norm of $\boldsymbol{\omega}$, i.e. $\boldsymbol{\omega}^\top \boldsymbol{\omega}$.

Table 9: Comparison of $\Delta m$ and performance variance for different methods on the NYUv2 dataset.

| method | LS | SI | RLW 2021a | DWA 2019 | UW 2018a |
|---|---|---|---|---|---|
| $\Delta m$ | 5.59 | 4.39 | 7.78 | 3.57 | 4.05 |
| $\mathrm{Var}[\Delta m_i]$ | 259.13 | 247.77 | 205.32 | 191.93 | 190.73 |

| method | MGDA 2018 | PCGrad 2020a | GradDrop 2020 | CAGrad 2021a | IMTL-G 2021b |
|---|---|---|---|---|---|
| $\Delta m$ | 1.38 | 3.97 | 3.58 | 0.20 | -0.76 |
| $\mathrm{Var}[\Delta m_i]$ | 68.65 | 173.67 | 204.45 | 137.94 | 124.03 |

| method | Moco 2023 | Nash-MTL 2022 | FAMO 2024 | FairGrad 2024 | **PIVRG (Ours)** |
|---|---|---|---|---|---|
| $\Delta m$ | 0.16 | -4.04 | -4.10 | -4.66 | **-6.50** |
| $\mathrm{Var}[\Delta m_i]$ | 163.07 | 108.03 | 74.66 | 71.59 | **52.21** |

Table 10: Comparison of $\Delta m$ and performance variance for different methods on the QM9 dataset.

| method | LS | SI | RLW 2021a | DWA 2019 | UW 2018a |
|---|---|---|---|---|---|
| $\Delta m$ | 177.6 | 77.8 | 203.8 | 175.3 | 108.0 |
| $\mathrm{Var}[\Delta m_i]$ | 59317.63 | 11807.60 | 77380.19 | 56660.16 | 18171.92 |

| method | MGDA 2018 | PCGrad 2020a | CAGrad 2021a | IMTL-G 2021b | Nash-MTL 2022 |
|---|---|---|---|---|---|
| $\Delta m$ | 120.5 | 125.7 | 112.8 | 77.2 | 62.0 |
| $\mathrm{Var}[\Delta m_i]$ | 20533.84 | 31570.73 | 18343.53 | 3309.90 | 10385.12 |

| method | FAMO 2024 | FairGrad 2024 | **PIVRG (Ours)** |
|---|---|---|---|
| $\Delta m$ | 58.5 | 57.9 | **33.6** |
| $\mathrm{Var}[\Delta m_i]$ | 3963.84 | 7705.27 | **3196.32** |

Table 11: Comparison of $\Delta m$ and performance variance for different methods on the Cityscapes dataset.

| method | LS | SI | RLW 2021a | DWA 2019 | UW 2018a |
|---|---|---|---|---|---|
| $\Delta m$ | 22.60 | 14.11 | 24.38 | 21.45 | 5.89 |
| $\mathrm{Var}[\Delta m_i]$ | 803.24 | 133.23 | 879.98 | 630.71 | 21.32 |

| method | MGDA 2018 | PCGrad 2020a | GradDrop 2020 | CAGrad 2021a | IMTL-G 2021b |
|---|---|---|---|---|---|
| $\Delta m$ | 44.14 | 18.29 | 23.73 | 11.64 | 11.10 |
| $\mathrm{Var}[\Delta m_i]$ | 3588.05 | 466.50 | 871.26 | 220.86 | 261.86 |

| method | MoCo 2023 | Nash-MTL 2022 | FAMO 2024 | FairGrad 2024 | **PIVRG (Ours)** |
|---|---|---|---|---|---|
| $\Delta m$ | 9.90 | 6.82 | 8.13 | 5.18 | **-0.54** |
| $\mathrm{Var}[\Delta m_i]$ | 126.75 | 128.77 | 73.43 | 53.26 | **1.55** |

## C  COMPARISON WITH OTHER METHODS

In this section, we present a concise overview of representative loss-based and gradient-based approaches used in multitask or multiobjective optimization, and provide a brief analysis of the characteristics of each method.

## C.1 Loss-Based Methods

**Linear scalarization (LS).** LS aims to directly optimize the average of all task losses. The optimization objective for LS is given by

$$\mathcal{L}(\theta) = \min_{\theta} \frac{1}{k} \sum_{i=1}^{k} \ell_i(\theta),$$

where $\ell_i(\theta)$ represents the loss for task $i$. LS focuses on minimizing the overall average loss, treating each task equally without considering individual task difficulties or imbalances.

**Scale-Invariant (SI).** The SI method aims to optimize the logarithmic mean of all task losses. The optimization objective for SI is given by

$$\min_{\theta} \frac{1}{k} \sum_{i=1}^{k} \log(\ell_i(\theta)),$$

where $\ell_i(\theta)$ represents the loss for task $i$. The advantage of SI is that it is invariant to any scalar multiplication of task losses, allowing it to handle varying loss scales effectively.

**Dynamic Weight Average (DWA) Liu et al. (2019).** It is a heuristic for adjusting task weights based on rates of loss changes. The optimization objective is a weighted sum of all task losses, where the weights are $\lambda_i$:

$$\min_{\theta} \sum_{i=1}^{k} \lambda_i \ell_i(\theta).$$

Similar to PIVRG, it also uses a softmax with temperature to determine the weights such that they sum to $k$. However, the softmax argument is $w_{i,t} = \ell_{i,t}/\ell_{i,t-1}$, which considers the relative change at the loss-level.

**Random Loss Weighting (RLW) Lin et al. (2021a).** The optimization objective of RLW is also a weighted sum of all task losses, where the weights are $\lambda_i$:

$$\min_{\theta} \sum_{i=1}^{k} \lambda_i \ell_i(\theta).$$

Unlike previous methods, RLW simply samples from a normal distribution and applies softmax to obtain the weights. The authors found that even this simple modification leads to better performance. They argue that RLW provides a higher probability of escaping local minima compared to existing models with fixed task weights, resulting in improved generalization ability.

**Fast Adaptive Multitask Optimization (FAMO) Liu et al. (2024).** FAMO aims to decrease all task losses at an equal rate at each step as much as possible. The optimization objective is:

$$\max_{d \in \mathbb{R}^n} \min_{i \in [k]} \frac{\ell_{i,t} - \ell_{i,t+1}}{\eta_t \ell_{i,t}} - \frac{1}{2}\|d_t\|^2,$$

where $\eta_t$ is the current step size. By amortizing over time, the authors propose a fast approximation to the solution, thus achieving highly competitive results while maintaining efficiency.

## C.2 Gradient-Based Methods

**Multiple Gradient Descent Algorithm (MGDA) Sener & Koltun (2018).** The MGDA algorithm is one of the earliest gradient manipulation methods for multitask learning. In MGDA, the per step update $d_t$ is found by solving

$$\max_{d \in \mathbb{R}^n} \min_{i \in [k]} g_{i,t}^{\top} d - \frac{1}{2}\|d\|^2.$$

As a result, the solution $d^*$ of MGDA optimizes the worst improvement across all tasks or equivalently seeks an equal descent across all task losses as much as possible. However, in practical applications, MGDA often encounters slow convergence due to the potential for $d^*$ to be quite small. For instance, if one task has a very small loss scale, the advancement of other tasks becomes constrained by the progress made on this particular task.

**Projecting Gradient Descent (PCGrad) Yu et al. (2020b).** PCGrad initializes $v_{\text{PC}}^i = g_{i,t}$, then for each task $i$, PCGrad loops over all task $j \neq i$ (in a random order, which is crucial as mentioned in Yu et al. (2020b) and removes the "conflict"

$$v_{\text{PC}}^i \leftarrow v_{\text{PC}}^i - \frac{v_{\text{PC}}^{i^\top} g_{j,t}}{\|\ell_{j,t}\|^2} g_{j,t} \quad \text{if} \quad v_{\text{PC}}^{i^\top} g_{j,t} < 0.$$

In the end, PCGrad produces $d_t = \frac{1}{k} \sum_{i=1}^k v_{\text{PC}}^i$. Due to the construction, PCGrad will also help improve the "worst improvement" across all tasks since the "conflicts" have been removed. However, due to the stochastic iterative procedural of this algorithm, it is hard to understand PCGrad from a first principle approach.

**Conflict-averse Gradient Descent (CAGrad) Liu et al. (2021a).** In CAGrad, $d_t$ is found by solving

$$\max_{d \in \mathbb{R}^m} \min_{i \in [k]} g_{i,t}^\top d \quad \text{s.t.} \quad \|d - \nabla \ell_{0,t}\| \leq c \|\nabla \ell_{0,t}\|,$$

where $\ell_{0,t} = \frac{1}{k} \sum_{i=1}^k \ell_{i,t}$. CAGrad aims to determine an update $d_t$ that maximizes the "worst improvement" while ensuring that the overall average loss decreases. By adjusting the hyperparameter $c$, CAGrad can replicate the behavior of MGDA when $c \to \infty$ and revert to the standard averaged gradient descent when $c \to 0$.

**Impartial Multi-Task Learning (IMTL-G) Liu et al. (2021b).** IMTL-G finds $d_t$ such that it shares the same cosine similarity with any task gradients:

$$\forall i \neq j, \quad d_t^\top \frac{g_{i,t}}{\|g_{i,t}\|} = d_t^\top \frac{g_{j,t}}{\|g_{j,t}\|}, \quad \text{and} \quad d_t = \sum_{i=1}^k w_{i,t} g_{i,t}, \text{ for some } w_t \in \mathbb{S}_k.$$

The constraint that $d_t = \sum_{i=1}^k w_{i,t} g_{i,t}$ is for preventing the problem from being under-determined. We can view IMTL-G as the equal angle descent, which is also proposed in Katrutsa et al. (2020), where the objective is to find $d$ such that

$$\forall i \neq j, \qquad \cos(d, g_{i,t}) = \cos(d, g_{j,t}).$$

**Nash-MTL Navon et al. (2022).** Nash-MTL finds $d_t$ by solving a bargaining game treating the local improvement of each task loss as the utility for each task:

$$\max_{d_t \in \mathbb{R}^n, \|d_t\| \leq \epsilon^2} \sum_{i=1}^k \log \left( g_{i,t}^\top d_t \right).$$

Note that the objective of Nash-MTL implicitly assumes that there exists $d_t$ such that $\forall i, \ g_{i,t}^\top d_t > 0$, otherwise we reach the Pareto front. In our proposed PIVRG, we also adopt this assumption.

$\alpha$**-Fair Resource Allocation (FairGrad) Ban & Ji (2024).** FairGrad is inspired by fair resource allocation in communication networks. They treat the optimization in MTL as a resource allocation problem and apply the $\alpha$-fairness framework:

$$U_\alpha(d) = \begin{cases} \sum_{i=1}^k \frac{u_i(d)^{1-\alpha}}{1-\alpha} & \text{if } \alpha > 0, \alpha \neq 1 \\ \sum_{i=1}^k \log(u_i(d)) & \text{if } \alpha = 1 \end{cases}$$

They also consider $g_i^\top d$ as the utility of task $i$. By introducing the $\alpha$-fair framework, FairGrad achieves different types of fairness at the gradient level, yielding surprising results. It is noteworthy that most existing methods can also be categorized under the $\alpha$-fair framework. For instance, LS is a special case when $\alpha = 0$, Nash-MTL corresponds to $\alpha = 1$, and MGDA is a special case as $\alpha$ approaches infinity. Similar to these methods, our basic optimization objective in Eq. 2 can also be viewed as a special case of $\alpha$-fairness. However, our derivation is from the perspective of minimizing the average optimization steps for tasks, and this is not our main contribution.

## C.3 Advantages of Our Method

Through the analysis of the aforementioned methods, we found that since loss-based methods cannot obtain the accurate gradient for each task, they primarily achieve fairness at the loss level through various scaling and weighted averaging of the loss. A major idea of gradient-based methods is to alleviate gradient conflict during the optimization process to achieve fairness at the gradient level. Additionally, some gradient-based methods use the first-order Taylor expansion to design utility functions, approximating the loss difference with $g_i^\top d$, thereby incorporating loss-level information.

However, only our proposed PIVRG considers performance-level information and uses the variance of performance drop as a fairness indicator to redefine fairness in the optimization process of MTL. Extensive experiments demonstrate that PIVRG not only achieves state-of-the-art performance but also realizes further fair optimization, mitigating the common task imbalance phenomenon observed in previous methods. Integrating our dynamically designed weighting strategy based on performance-level information into existing methods can significantly enhance their performance and reduce the variance of performance drop, achieving more equitable results. This further confirms the potential of our method and its contribution to the MTL community.

