# OpenReview forum: "Rethinking Fairness Representation in Multi-Task Learning: a Performance-Informed Variance Reduction Approach"
_ICLR.cc/2025/Conference — ICLR 2025 Conference Withdrawn Submission_

### Official Review · Reviewer_yJJs · 2024-10-28

**Soundness:** 2
**Presentation:** 3
**Contribution:** 2
**Rating:** 3
**Confidence:** 5

**Summary:**

This paper utilizes performance variance across multiple tasks as additional information to guide the training process, and then proposes a performance-informed variance reduction gradient (PIVRG) method for multi-task learning. Theoretical analysis shows that PIVRG can converge to a Pareto stationary point under certain assumptions. Extensive empirical studies further demonstrate its effectiveness. Additionally, the performance-informed idea can also be applied to other existing MTL methods.

**Strengths:**

The proposed PIVRG outperforms all baselines on many multi-task scenarios, including those for supervised learning and reinforcement learning.

**Weaknesses:**

1. The novelty is insufficient. Since the paper compares with FairGrad[1], it appears that Eq. (2) in this paper corresponds to minimum potential delay fairness, a specific case of FairGrad. Additionally, the method used to solve Eq. (8) is the same as in FairGrad.

2. The use of $\Delta m$ to guide the training process seems kind of strange. Although the paper mentions in Sec 4.1 that $\Delta m$ is derived from the validation dataset for QM9 and CelebA, and from the training dataset for NYU-v2 and Cityscapes, further clarification on the calculation would better be provided.\
From my understanding, $M_{b,i}$ in Eq. (3) denotes the metric value of the STL baseline, obtained from the test dataset. That is, when calculating $\Delta m$, information from the test dataset is involved. This type of **test** information should not be used to guide the **training** process, as it is also used to evaluate performance. Could you please more clarifications on the use of $\Delta m$?

[1] Ban, Hao, and Kaiyi Ji. "Fair Resource Allocation in Multi-Task Learning." arXiv preprint arXiv:2402.15638 (2024).

**Questions:**

For the reinforcement learning experiment on MT10, I noticed in Sec 5.5 of FairGrad that the authors found it very time-consuming to solve the nonlinear least square problem, so they approximated the objective using SGD in practice. Since you also treat Eq. (8) as a nonlinear least square problem, and appear to solve it directly, how long do the MT10 experiments take?

---

### Official Review · Reviewer_djmC · 2024-11-01

**Soundness:** 3
**Presentation:** 3
**Contribution:** 2
**Rating:** 6
**Confidence:** 4

**Summary:**

The paper proposes a novel method named PIVRG for MTL that alleviates the issue of task imbalance by incorporating performance-level information into the optimization process. The method introduces a dynamic weighting strategy that uses the variance of performance drop across tasks as a fairness indicator, aiming to reduce the performance variance and achieve more balanced optimization. Extensive experiments show that PIVRG outperforms existing methods and reduces performance variance.

**Strengths:**

1. Novelty: The paper introduces a new approach to MTL that minimizes the mean of the inverse utilities for each task and explicitly considers performance-level information for dynamic weighting in the optimization process.
2. Theoretical foundation: The authors provide a theoretical analysis demonstrating that PIVRG converges to a Pareto stationary point and achieves superior performance.
3. Strong performance: The paper includes comprehensive experiments on various benchmarks showing that PIVRG outperforms existing methods and reduces performance variance.

**Weaknesses:**

1. Incorrect definition of the $\Delta m$: The $\Delta m$ used in previous works and reported in the experiments of this work is different from the $\Delta m_i$ defined in Eq. 3 and the $\Delta m$ defined in Line 190. The former is calculated across all specific metrics, while the latter is computed in a single task and then averaged across all tasks. It is recommended that the performance-related calculation in Sec. 3 be redefined as a new indicator to avoid confusion with $\Delta m$. Additionally, the correct definition of $\Delta m$ should be clarified in Section 4.
2. Lack of discussion on some related work: [1] utilizes Key Performance Indicators (KPI) to dynamically prioritize diﬃcult tasks, which is also performance-level information. [2] focuses on improvable gap balancing across tasks, similarly prioritizing more difficult tasks. This is particularly comparable to the characteristics of PIVRG, as seen in the experimental results where the surface normal prediction performance on NYUv2 improves.
3. Lack of motivations and ablation studies on gradient aggregation approach of PIVRG: If Eq. 2 is one of the innovations of this paper, it would be beneficial to elaborate on its motivation and design process in the introduction and method sections. If it is not, I suggest including more ablation studies, such as comparisons like “PIVRG w/o PI”, and/or “Eq. 2 + other SOTA weighting methods”, to further validate the effectiveness of PI.

[1] Dynamic Task Prioritization for Multitask Learning. ECCV, 2018.

[2] Improvable Gap Balancing for Multi-Task Learning. UAI, 2023.

**Questions:**

Please my questions in Weaknesses.

---

### Official Review · Reviewer_66Es · 2024-11-04

**Soundness:** 3
**Presentation:** 3
**Contribution:** 1
**Rating:** 3
**Confidence:** 4

**Summary:**

The paper introduces a novel performance-informed variance reduction gradient aggregation approach (PIVRG) for multi-task optimization. This method uses performance-level information as an explicit fairness indicator and demonstrates its effectiveness across various benchmarks, enhancing multi-task performance.

**Strengths:**

1. The proposed motivation, which incorporates performance-level information, is intuitive, and the paper’s explanation of how it applies to multi-task optimization is clear and logical.

2. The paper thoughtfully follows the standard demonstration logic in the multi-task optimization field, providing necessary theoretical analysis based on widely accepted assumptions in this area.

3. The proposed method demonstrates improved multi-task performance across various benchmarks.

**Weaknesses:**

1. Using performance-level information is not a novel approach, as [1] already incorporates this concept for multi-task optimization through task difficulty, which the authors have cited in related work. Since this paper’s motivation is closely related to [1], further experiments and comparative analysis—including [1], which is currently omitted from the experiments—are needed to highlight distinctions.

[1] Guo, Michelle, et al. "Dynamic Task Prioritization for Multitask Learning." Proceedings of the European Conference on Computer Vision (ECCV), 2018.

2. TTechnically, the paper effectively incorporates performance-level information into multi-task optimization; however, it seems to be a naive combination of task weighting based on performance metrics with existing optimization approaches, offering limited new insights for the multi-task optimization field. The authors assert that their performance-informed weighting strategy can integrate with prior loss-based and gradient-based methods, as shown in Table 5. This raises concerns that the proposed method might simply be a basic combination of previous techniques, as performance gains in each approach can already be achieved by weighting based on evaluation metrics. For a fair comparison, experiments should include combinations of [1] with other methods, encompassing both loss-based and gradient-based approaches. This would help justify the superiority of the proposed methods in terms of their methodology for incorporating performance-level information.

3. The assumption that the network has access to performance-level information during optimization is quite strong. This limitation impacts the practicality of the proposed methods, particularly given that creating multi-task benchmarks is both costly and challenging due to the extensive labeling process required for multiple tasks. Additionally, the proposed methods necessitate extra validation or test sets for training, further restricting their applicability. This may explain why many previous studies have been conducted in experimental settings that do not rely on performance-level information.

**Questions:**

Refer to the weaknesses section.

---

### Official Review · Reviewer_UYd6 · 2024-11-04

**Soundness:** 3
**Presentation:** 3
**Contribution:** 3
**Rating:** 6
**Confidence:** 3

**Summary:**

- Paper identifies task imbalance, one of the common issues in multi-task learning (MTL) that leads to inefficient learning dynamics to address.
- The paper opines that the existing methods of loss-based and gradient-based imbalances lead to uneven optimization and limiting the generalization capability of the model.
- The paper introduces a fairness-driven approach that dynamically balances task optimization to address task, loss, and gradient imbalances.
- The proposed method (PIVRG) uses the variance in performance $(Δm)$ across tasks as a fairness indicator and implements a dynamic weighting mechanism to progressively reduce variance among tasks, ensuring balanced optimization.
- The effectiveness of the proposed approach is validated through comprehensive experiments involving both supervised and reinforcement learning tasks.
- The results demonstrate state-of-the-art performance, highlighting the superiority (reducing the performance variance across tasks) of the proposed method over traditional approaches.

**Strengths:**

1. Fairness Regularizer: The paper introduces a performance metric as a fairness measure in terms of a regularizer in the backpropagation algorithm.
2. Rigorous Experimentation: The paper presents rigorous experimentation with strong results across various datasets.
3. Better Performance: Results demonstrate that the proposed method consistently performs better in their multi-task learning (MTL) setup than in single-task learning.

**Weaknesses:**

- Please refer to the 'Questions' section of the review.

**Questions:**

1. Performance Metric for $Δm$: What specific performance metric is employed for computing $Δm$? Some examples related to datasets involving a smaller number of tasks, such as NYU-v2, would be useful for further understanding.

2. Source of $Δm$: It is mentioned that $Δm$ is obtained from the validation dataset ($D_v$). Why is it not obtained from the test dataset $(D_t)$? This aspect is not clearly explained in the paper.

3. Performance Dependency on Task Complexity: Does the performance metric vary based on the complexity of each task? Should the choice of performance metric not account for the specific relevance of the task? Would it be beneficial to consider different performance metrics for each task depending on the task's nature?

4. Direction Variable (d) in the Utility Term: There seems to be ambiguity regarding the direction variable (d), which is task-specific. In the PIVRG algorithm, “d” is computed for shared layers based on task-specific variables. Is the direction variable for the shared layer identical to the task-specific direction variable? If so, in Equation 7, it appears that “d” is determined using a utility term that also contains “d”. Are these two variables distinct, or is this an iterative dependence that needs clarification?

5. Equation 8 Clarification: In Equation 8, it is not clear why there is an alpha on the LHS.

---

### Note · Authors · 2024-11-15

**Comment:**

We sincerely thank the reviewers for the time and effort spent on reviewing our manuscript and for providing valuable suggestions. Your comments have motivated us to refine this work further and continue improving its quality. Once again, we appreciate your suggestions.

**Withdrawal Confirmation:**

I have read and agree with the venue's withdrawal policy on behalf of myself and my co-authors.